# Electrical and Mechanical Characterisation of Poly(ethylene)oxide-Polysulfone Blend for Composite Structural Lithium Batteries

**DOI:** 10.3390/polym15112581

**Published:** 2023-06-05

**Authors:** Francesco Gucci, Marzio Grasso, Stefano Russo, Glenn J. T. Leighton, Christopher Shaw, James Brighton

**Affiliations:** School of Aerospace, Transport and Manufacturing, Cranfield University, Cranfield MK43 0AL, UK

**Keywords:** polymer blend, mechanical properties, multifunctional material, composite structural battery, polyethylene oxide

## Abstract

In this work, a blend of PEO, polysulfone (PSF), and lithium bis(trifluoromethanesulfonyl)imide (LiTFSi) was prepared at different PEO–PSf weight ratios (70-30, 80-20, and 90-10) and ethylene oxide to lithium (EO/Li) ratios (16/1, 20/1, 30/1, and 50/1). The samples were characterised using FT-IR, DSC, and XRD. Young’s modulus and tensile strength were evaluated at room temperature with micro-tensile testing. The ionic conductivity was measured between 5 °C and 45 °C through electrochemical impedance spectroscopy (EIS). The samples with a ratio of PEO and PSf equal to 70-30 and EO/Li ratio equal to 16/1 have the highest conductivity (1.91 × 10^−4^ S/cm) at 25 °C, while the PEO–PSf 80-20 EO/Li = 50/1 have the highest averaged Young’s modulus of about 1.5 GPa at 25 °C. The configuration with a good balance between electrical and mechanical properties is the PEO–PSf 70-30 EO/Li = 30/1, which has a conductivity of 1.17 × 10^−4^ S/cm and a Young’s modulus of 800 MPa, both measured at 25 °C. It was also found that increasing the EO/Li ratio to 16/1 dramatically affects the mechanical properties of the samples with them showing extreme embrittlement.

## 1. Introduction

Minimising catastrophic consequences of climate change is a significant driver for enabling energy transition and reducing dependency on fossil fuels. Electric vehicles and green energy produced by renewables provide focal points to address these global issues. Battery electric vehicles (BEVs) have, on average, 25% of the total weight taken up by batteries. This results in a higher total weight than the traditional combustion-engine car. There is an increasing demand for innovative solutions that can reduce the dead weight represented by conventional battery technologies, considered ‘structurally inefficient’, and exploit the potential of more readily available materials with a reduced impact on the environment.

To increase the capabilities to store and deliver energy, several strategies have been followed for improving the current lithium technology by allowing the use of Li-metal [1], Si-anodes [2], solid-state electrolytes [3,4], or high-voltage cathodes [5]. Additional research has also investigated materials to exploit alternative chemicals, such as sodium [6] or potassium [7]. All these fields are aimed at addressing a different issue related to the battery. For example, new electrodes aim to increase capacity and/or life, whilst solid electrolyte allows safer batteries since they are not flammable, such as the liquid ones, and can allow larger voltage windows. The development of multifunctional energy storage systems, i.e., where the battery may be structurally designed to support/contain other functionality, such as sensors, leads to the need to look at improving their mechanical requirements above that of a monofunctional solid-state battery. The combined properties required are dependent upon the targeted application and this influences not only the electrolyte but also the other components. Composite structural batteries [8] have been proposed to substitute for vehicle parts, such as rooftops, so they need to combine the energy-storage capacity with sufficient mechanical strength to withstand related stress. For this reason, one or both electrodes are carbon-fibre-based and the overall assembly is expected to approach the mechanical properties of resin composite. In contrast for wearable applications, the expectation is on achieving flexibility to adapt to complex surfaces. In the Li-S system, for example [9], graphene sheet-based electrodes have been extensively researched. For both applications, composite polymer electrolytes are extensively researched [8,9], since polymers can exhibit different mechanical behaviours. 

The most commonly studied polymer for electrolyte applications is polyethylene oxide (PEO), as it can dissolve alkali metal salts and achieve good ionic conductivity at high temperatures [10], although it has low mechanical properties. The reported mechanical strength of pure PEO varies from about 1 MPa to 15 MPa [11,12,13], dependent on its molecular weight and manufacturing route, with a very low conductivity value of around 10^−8^ S/cm. One strategy to improve the conductivity of the polymeric electrolyte is the addition of fillers, which can be inactive, such as SiO_2_ [14], Al_2_O_3_ [15], or graphene oxide (GO) [16], or ceramic ionic conductors, such as Li_10_GeP_2_S_12_ [17] or Li_1.5_Al_0.5_Ge_1.5_(PO_4_)_3_ [18]. Typically, the reported effects of the additives are connected to the electrochemical performances and less often to the mechanical properties. For example, Wen et al. [16] prepared a PEO–LiTFSI electrolyte with an ethylene oxide to lithium ratio (EO/Li) of 10:1 and used GO as a filler. An increase in the tensile strength from about 0.2 MPa to a maximum value of 1.31 MPa was observed with the addition of 1% in weight of GO. The conductivity was reported to have values of around 1.5 × 10^−6^ S/cm, which were seven times higher than the value for the unfilled composition. However, further increase in the concentration of GO resulted in a decrease in conductivity. Li et al. [19] prepared a PEO–LiTFSI electrolyte with an EO/Li ratio of 20:1 using a high molecular weight PEO and explored the effects of different concentrations of MnO_2_ nanosheet. Samples prepared with 5% in weight of filler showed a maximum strength of about 2 MPa, twice that of the unfilled composition, while the conductivity values at room temperature (RT) were 1.95 × 10^−5^ S/cm for the samples with filler and 1.38 × 10^−5^ S/cm for the PEO–LiTFSI. Lee et al. [20] used a lower molecular weight PEO with the same EO/Li ratio and Li_1.4_Al_0.4_Ge_1.6_(PO_4_)_3_ (LAGP) as a filler. The addition of 20% in weight of LAGP caused a small increase in the conductivity values at room temperature, from 4 × 10^−6^ S/cm to 5 × 10^−6^ S/cm, and an increase in the tensile strength from 2.0 MPa to 2.5 MPa. A further increase in the amount of filler beyond 20% in weight lowered the conductivity and increased brittleness with similar tensile strength. 

Blending PEO with other polymers and plasticizers is another strategy used to modify the electrolyte properties [21], even though the effect on mechanical properties is rarely reported. Tao et al. [22] manufactured a blend of different weight ratios of PEO and thermoplastic polyurethane (TPU) with a concentration of LiTFSI fixed at 30% in weight. The PEO alone with LiTFSI showed a tensile strength of 0.18 MPa, while TPU alone with LITFSI achieved 17.31 MPa, and subsequent polymericblends achieved values between these two extremes. Similarly, the conductivity at 20 °C was 10^−5^ S/cm for PEO and decreased with increasing amount of TPU down to about 6 × 10^−7^ S/cm for TPU alone. Xu et al. [23] mixed PEO and polyvinylidene-fluoride (PVDF) in a 70-30 weight ratio adding ZnO as filler and LiClO_4_ to achieve an EO/Li ratio of 10:1. A tensile strength of 2.35 MPa was achieved at 2% in weight of ZnO, while without the ZnO it reached 1.28 MPa. The conductivity at room temperature had a maximum value for 2% in weight at about 6 × 10^−5^ S/cm, but decreased with higher values, and for 5% in weight it was even lower than the composition without ZnO which had showed about 2.2 × 10^−5^ S/cm, highlighting the susceptibility of the polymer to small increases in the filler content. Complex copolymers have also been studied. For example, Zhang et al. [24] developed a pentablock copolymer composed of polystyrene (PS)–PEO–Polypropylene (PP)–PEO–PS and different percentages in weight of Li_7_La_3_Zr_2_O_12_ and LiTFSI for an EO/Li ratio of 11/8/1. The mechanical strength of the copolymer was 1.68 MPa, but the addition of 30% in weight of LLZO reduced the maximum strength to 0.92 MPa, whilst the conductivity at room temperature increased from about 1.5 × 10^−5^ S/cm to about 2 × 10^−4^ S/cm. For structural purposes, two system electrolytes have also been proposed. Schneider et al. [25] developed an electrolyte having a mechanically strong phase obtained from bisphenol A ethoxylate dimethacrylate and a conducting liquid phase based on dimethyl methylphosphonate, ethylene carbonate, and 1 M LiTFSI. The weight ratio of the liquid phase in the original mixture was 39%, and the room-temperature storage modulus was around 530 MPa, regardless of the curing temperature or method (UV or thermal), while the conductivity was about 2 × 10^−4^ S/cm. One class of polymers that has gained attention as part of the electrolyte are polysulphones, which are commonly used for their chemical and mechanical stability. Lu et al. produced a copolymer between PEO and polysulfone (PSf) with a PEO to LiTFSI ratio of 8:1 and added succinotrile (SN) to investigate its effect [26]. The best performances were achieved for a 35% in weight of PEO, with a conductivity at room temperature of 10^−7^ S/cm at 30 °C and a tensile strength of 14.7 MPa. When SN was added to achieve a PSF–PEO: SN: LiTFSI weight ratio 50/35/15, the conductivity increased to 10^−4^ S/cm, while the tensile strength decreased to 9.6 MPa. Xu et al. [27] prepared a composite electrolyte from PEO–polyoxyphenylene sulphone (PESf)–polyvinylacetate (PVA) and LiTFSI for a ratio of EO/Li = 16/1. The best composition having 30% in weight of PVA and 20% in weight PESf had a room-temperature conductivity in the order of 10^−5^ S/cm, but mechanical properties were not reported. 

In this work, we investigate the mechanical and electrochemical properties of a PEO–PSf blend with a fixed amount of LiTFSI and the effect of LiTFSI on a PSf-rich composition, which have not been reported before. Additionally, the films are simply cast from dimethylsulfoxide (DMSO) instead of acetonitrile or tetrahydrofuran which are more commonly used but are toxic. The developed system involves a simple manufacturing process and a relatively low salt concentration, yet possesses good properties compared to other systems developed in the literature and confirms the potential of sulfone-based polymers as battery electrolytes.

## 2. Materials and Methods

### 2.1. Materials

The materials used in this work were polyethylene oxide (molecular weight 4 × 10^5^ g/mol, Sigma Aldrich, Gillingham, UK), dimethylsulfoxide (99% pure, Sigma Aldrich, Gillingham, UK), polysulfone (molecular weight ~30,000–40,000 g/mol, Sigma Aldrich, Gillingham, UK), and lithium bis(trifluoromethane)sulfonimide (99% pure, Sigma Aldrich, Gillingham, UK). The PEO was dried overnight in a vacuum oven at 50 °C, while PSf and LiTFSI were dried at 110 °C. The preparation of the solutions is similar to what was reported in our previous work [28], with the components being dissolved in an appropriate amount of DMSO and left to stir for 48 h. The PEO–PSF weight ratios were initially 90-10, 80-20, and 70-30 with an EO/Li ratio of 50/1. Other samples based on the 70-30 PEO–PSF ratio were manufactured with EO/Li ratios of 30/1, 20/1, and 16/1. The solutions were then placed in a plasma-cleaned Teflon mould and dried in a vacuum oven at 60 °C for 48 h to obtain a self-supporting continuous film with, typically, 0.2 mm thickness and 50 × 50 mm area. For clarity, the sample code names have been expressed in the following format shown in Table 1. 

### 2.2. Sample Characterisation

To characterise the composite films, a number of analytical techniques were carried out. X-ray diffraction (XRD) analysis was performed on a Siemens/Bruker D5000 (Siemens, Munich, Germany) to determine the degree of crystallinity using the relationship shown in Equation (1): (1)$$X=\frac{I_{c}}{I_{t}}$$
where X is the crystallinity degree, $I_{c}$ is the intensity of the crystalline peaks, and $I_{t}$ is the total peak intensity. Fourier transformed infrared (FT-IR) analysis has been conducted on a Jasco FT-IR 6200 with a sensitivity of 4 cm^−1^ in the range 650–4000 cm^−1^.

Differential scanning calorimetry was performed using a TA Q200 with RCS90 as a cooler (TA Instruments, Wilmslow, UK), and the crystallinity of the PEO was calculated using Equation (2):(2)$$X_{C}=\frac{{\Delta H}_{e}}{{\Delta H}_{c}}$$
where ${\Delta H}_{c}$ is the enthalpy of the melting of crystalline PEO corresponding to 205 J/g [29] and ${\Delta H}_{e}$ is the recorded value. A heat–cool–heat cycle between −85 °C and 100 °C was set to perform DSC analysis, at a rate of 10 °C/min and holding the lowest and highest temperature for 5 min, using the method suggested by Ref. [30]. Micro-tensile testing was performed on a Deben micro-test tensile stage (Deben UK Ltd., Oxford, UK) with a 200 N cell. Micro-tensile tests were conducted according to ASTM D882 (“Standard Test Method for Tensile Properties of Thin Plastic Sheeting”) [31]. Samples were cut as strips of uniform width of 5 mm and length of 25 mm. Each sample tested was initially inspected using an optical microscope which was used to take some representative images and to check for visible flaws. While the most common method to observe polymeric crystals would be a cross-polarized microscope, it is possible to identify them also with a standard microscope [32]. The force and displacement values measured during the tensile tests, whilst applying a constant displacement of 1 mm/min, were converted into engineering stress and engineering strain through the relationships shown in (Equations (3) and (4)): (3)$$\sigma =\frac{F}{A}$$
(4)$$\epsilon =\frac{∆l}{l_{0}}$$
where σ is the stress expressed in MPa, *F* is the force expressed in newtons, *A* is the cross-section expressed in mm^2^, ϵ is strain, ∆*l* is the variation in length, and $l_{0}$ is the initial length of the sample, both expressed in mm. Electrochemical impedance spectroscopy was performed on a Gamry Interface 1010E potentiostat (Gamry Instruments, Philadelphia, PA, USA) in the range of 2 MHz to 1 Hz and the blocking electrode setup consisted of a 13 mm Swagelok-type cell (Cambridge Energy Solutions, Cambridge, UK), which was placed in an oven and connected to the Gamry Interface 1010E. To ensure thermal equilibrium, the Swagelok cell was left inside the oven for roughly 2 h at the same temperature. The temperature was constantly monitored through a RS1314 dual thermometer (RS Components Ltd., Corby, UK) in contact with the surface of the Swagelok-type cell. The conductivity of the electrolyte was calculated according to [33] using Equation (5):(5)$$\sigma =\frac{t}{AR}$$
where σ is the conductivity in S/cm, t and A are the thickness and surface area of the electrolyte, respectively, and R is the bulk resistance obtained from the impedance spectroscopy. The value of the bulk resistance is obtained by fitting the data using a circuit for non-ideal blocking electrode [34,35], as shown in Figure 1. *R_b_* represents the bulk resistance of the electrolyte, a constant phase element (*CPE_d_*) is used to represent the non-ideal behaviour of the geometrical capacitance of the solid polymer electrolyte (SPE)/electrode structure, and a constant phase element (*CPE_e_*) is used to represent the non-ideal behaviour of the electric double layer at the SPE/electrode interfaces. 

The contact angle has been measured using a theta lite instrument from Biolin Scientific (Manchester, UK) using the sessile-drop technique with deionized water droplets of about 1.5 µL. Following initial testing, there was evidence that the water droplets were leaving impressions in the surface of the samples, probably indicating partial dissolution of material within the time-frame of the experiment. Consequently, the measurements are an average of three values taken during the first few seconds of contact with the surface to try to minimise changes to the nature of the DI water. 

## 3. Results and Discussion

### 3.1. X-ray Diffraction Analysis

The XRD data collected for all samples are shown in Figure 2. Typical peaks of the crystalline phase of PEO are the single peak at 19 degrees and the triple peak at 24 degrees. The halo underneath is an indicator of an amorphous phase, since PEO is a semi-crystalline polymer [21]. Peaks associated with crystal complexes instead are not present suggesting that PSf is able to prevent their formation, detrimental on the overall ionic conductivity [36], as they have been observed below 20/1 EO/Li ratio [20,37]. While the crystallinity of PEO is near 70%, consistent with data reported by other authors [38], the addition of 10% in weight of PSf and 50/1 LiTFSI for the PS9150 samples brings it down to 61.8%, and this is further reduced by increasing PSf up to 30% in weight for the PS7350 samples which allows a crystallinity of 44.4%, as shown in Table 1. Decreasing the EO/Li ratio from 50/1 to 16/1 further reduced the crystallinity, but the effect is limited as the minimum value achieved was 35.6%.

### 3.2. Differential Scanning Calorimetry Analysis

The heat–cool–heat cycle is thought to cancel the thermal history of the samples, which depends on the manufacturing and storage of the films [39]. Therefore, the second heating cycle is taken as reference for the properties displayed in Table 2. 

In Figure 3, the normalised second heating scan of all the samples is shown with melting peaks and glass transition temperatures. The melting temperature value decreased as the concentration of PSf and LiTFSI increased with the exception of the PS7320 sample. Its behaviour suggested that a different interaction between the PEO–PSf–LiTFSI is produced at this composition. Data for melting point, glass transition temperature, enthalpy of melting, and crystallinity percentage are shown in Table 2. Also, crystallinity computed using Equation (1) is compared to the crystallinity obtained by XRD using Equation (2). 

The values measured for the glass transition temperature are very close to each other and to virgin PEO, while the PSf glass transition temperature is known to be at a higher temperature range. Since the effect of PSf is evident from the mechanical and electrical properties, the observed single glass transition temperature suggests that it does belong to a PEO-rich phase but there is at least a PSF-rich phase with a glass transition outside of the temperature range −85 °C to 100 °C.

Moreover, at close to room temperature, the environmental effect can significantly affect the measurements, and this can justify the discrepancy between the PS7316 and PS7330 samples. The discrepancy between the two crystallinity values is high but the peak crystallization for these compositions is 27 °C and 31 °C, respectively, which might have caused a crystallization over time that has affected the measurements [38].

### 3.3. Electrochemical Impedance Spectroscopy Analysis

The EIS curves taken at different temperatures for the PS7330 samples are shown in Figure 4. 

As commonly happens [40], the increase in temperature causes a reduction of the resistance and, therefore, an increase in conductivity. The trend is common to all sample configurations (Figure 5).

The conductivity at 25 °C measured in this work for the PS9150 samples (4.83 × 10^−5^ S/cm) is higher than reported in previous work [28], and, generally, considered a high value [21,41,42]. This could indicate that the longer dissolution time in the film-manufacturing approach was particularly beneficial. In comparison, this value was better than the ionic conductivity at 25 °C for both PS7350 samples (3.11 × 10^−5^ S/cm) and PS8250 samples (3.62 × 10^−6^ S/cm). The PS7330 and PS7316 samples with the high concentration of salt showed higher conductivity with values of 1.17 × 10^−4^ S/cm and 1.91 × 10^−4^ S/cm, respectively. The PS7320 samples showed lower conductivity values over the entire temperature range, compared with the PS7330 samples. This might be seen as counterintuitive, as a higher amount of salt should represent a higher ionic conductivity, but this specific composition did not follow a pseudo-linear trend in any of the other characterizations performed in this study. As the results from the DSC suggest, the samples’ morphology comprises several phases, where the amorphous phase(s) is responsible for most of the ionic conductivity. The lower conductivity of the PS7320 samples can, probably, be related to a different morphology, which allows for a different degree of crystallization. This, in turn, hinders the conductivity, as evidenced by the diverse interaction between the components shown by the FT-IR analysis which follows. From Figure 5, it is also worth noticing that at the temperature of 5 °C, which still represents an operating temperature for structural solid-state batteries, the ionic conductivity of the PS9150 samples averages at 1.88 × 10^−5^ S/cm, the PS7330 samples show values around 2.85 × 10^−5^ S/cm, and the PS7316 samples exhibit a remarkable 4.71 × 10^−5^ S/cm. Overall, the composition which demonstrates the highest conductivity for the whole temperature range is certainly that of the PS7316 samples. However, when considering the multidisciplinary approach being considered in this article, the mechanical testing results still need to be taken in tandem with the conductivity measurements.

### 3.4. Fourier-Transform Infrared Spectroscopy Analysis

The most important peaks to identify the organic components are in the range of 1800 cm^−1^ to 700 cm^−1^ (Figure 6b). The triplet peaks centred at 1090 cm^−1^ are associated with the crystalline phase of PEO [24,34], and appear slightly shifted with an increasing amount of PSf, suggesting some interaction between the two polymers. Similarly, when the LiTFSI salt content is increased, the relative ratio of the triplets is significantly modified, indicating that the salt is altering the crystalline phase, as observed in the data from the XRD and the DSC. The peak around 1193 cm^−1^ is associated with the CF_3_ stretching [43] and its intensity increases with the amount of salt.

The small peaks at 1359 cm^−1^ and 1340 cm^−1^ can be associated with the methylene group wagging (-CH_3_) in the crystalline phase and tend to merge and broaden with the increasing amount of PSf and LiTFSI. This could suggest an interaction between these components and the crystalline phase of PEO. The low-intensity peak at 865 cm^−1^ is related to the benzene ring [27] of the PSf, but does not appear to be influenced by the overall composition of the blend.

The peak at 1240 cm^−1^ and 1280 cm^−1^ can be associated with the twisting of the CH_2_ bond [44], and the variation of their relative ratio can be associated with a different orientation of the PEO chains [45]. This is increasingly modified with higher amounts of PSf and LiTFSI, and it can be related to the superior mechanical properties of the blends with 50/1 EO/Li ratio. The peaks at 840–850 cm^−1^, and, to a lesser extent, at 2800 cm^−1^ (Figure 6a), are slightly modified by the addition of a high amount of LiTFSI. This indicates a complexation, as shown by Wen et al. [46], even if for a relatively low amount of salt as in 16/1 EO/Li ratio the variation is limited. Additionally, the broad peak above 3000 cm^−1^ can be associated with water absorption, as the salt has the tendency to absorb it from the environment much more readily than PEO. Another couple of peaks of interest are placed at 1030–1070 cm^−1^ which can be related to the sulfonyl group SO_2_ [47], but also overlap with PEO. The increasing amount of PSf and then LiTFSI causes a change in the overall shape of the absorption curve, with a marked increase of the latter for the PS7330 composition and PS7316 composition, for which it is higher than the PEO crystalline peaks, and less intense for PS7320 composition. This suggests that the PSf or LiTFSI (as both possess SO_2_) have a different interaction at this concentration, which could relate to the different mechanical properties. The PS7320 samples have a higher maximum stress, while the PS7316 ones display brittle fractures, and this might be due to the strong interfacial bond responsible for the separation of the amorphous phase from the crystalline phase. A summary of peak wavenumbers and vibration modes is shown in Table 3. 

### 3.5. Micro-Tensile Testing

The micro-tensile testing data recorded during the test are force and displacement. The nominal stress is calculated using the initial area of the specimens obtained as an average of several values of thickness and width measured along the length of the specimen. Before each test, the specimens were observed with optical microscopy to identify the typical distribution of the crystals and ratio between the amorphous and the crystalline phases. In order to assess the effects of the PSf and the Li salt, a number of pure PEO samples were tested as reference. The average stress-elongation curve obtained for the pure PEO is shown in Figure 7a. The polymer shows a linear elastic behaviour up to the maximum stress recorded (over 10 MPa), and, then, the corresponding number and extension of the voids in the amorphous phase are responsible for the observed drop in the stress-elongation curve from 0.25 mm. The sample after the peak stress can sustain a continuous deformation, with a constant stress around 8 MPa until the extension of the voids significantly reduce the strength of the film and the stress drops to zero. 

Following the trend observed in the previous study [28], the main focus in this work was on the PEO–PSf 70-30 ratio compositions, which had shown the best mechanical properties in that study. One immediate observation was that the small variation in the manufacturing procedure had a significant impact on the mechanical properties. The PEO (Figure 7a) achieved a maximum stress of 10 MPa ± 3.3 MPa and a Young’s modulus of 690 MPa ± 140 MPa. The literature reports different values for PEO [48,49,50]. The addition of LiTFSI or other salts is known to reduce the mechanical properties of PEO [20], and it has been observed that thickness variation can have an impact on mechanical properties [51,52], and large fluctuations are also observed in salt-containing blends [53].

The average stress-elongation curves for the composite samples are shown in Figure 7b–f, with a macroscopic image taken at 1.5 mm elongation for all the samples except for PS7316, for which rupture had occurred at around 0.4 mm. A summary of the average Young’s modulus and maximum stress values, which are obtained from the stress-elongation curves, are in Figure 8a,b. Figure 7b displays the curve for the PS9150 samples, which appears to have a mostly ductile behaviour, as the curve has a steep onset and then stabilises around the stress value of 17.5 MPa. A similar trend is shown by the PS8250 samples (Figure 7c), which also reaches a peak at around 17 MPa, before decreasing and stabilising at around 14 MPa. Figure 7d shows the stress-elongation curve for the PS7350 samples. In this case, the samples displayed slight shifting towards a brittle behaviour, as also confirmed by the image taken during the test and shown underneath the curve. Nonetheless, the samples in Figure 7d still manage to sustain up to about 16 MPa of stress. Figure 7e displays the averaged stress-elongation curve for the PS7330 samples, which also appears to have a somewhat brittle nature, reaching a peak at around 10 MPa and then displaying a slow decrease in the stress handling. As indicated by the previous characterization methods, the PS7320 samples once again behave in an ambiguous way compared to the PS7330 and PS7316 compositions. Figure 7f shows the averaged stress-elongation curve for the PS7320 samples, revealing a behaviour similar to the PEO–PS9150 samples, as the curve steeply reaches and holds about 13 MPa of stress. This is also confirmed by the image of the sample shown in Figure 7f, as the sample itself is deformed in a dissimilar way compared to the sample’s images in Figure 7e,g.

Finally, the curve for the PS7316 samples is shown in Figure 7g where, as expected, the behaviour is clearly brittle, and the samples show little elastic deformation and failure after a limited elongation. A closeup view of the fracture area of a PS7316 sample is shown in Figure 9c. 

The Young’s modulus of PSf is around 2.5–3 GPa [54,55], while the PEO blends containing 10 and 20% in weight PSf are above 1 GPa (Figure 8a) and the modulus is still around 800 MPa for PS7350, PS7330, and PS7320. The minimal impact of the salt addition on the mechanical properties seems valid until the salt content is high enough to dramatically influence the modulus, as seen for PS7316, where the value is below 200 MPa. The maximum stress follows a similar trend, as the addition of 10% in weight PSf increases the value from about 10 MPa to close to 19 MPa, but a further increment of 20 and 30% in weight reduces this value to 17.5 and 16 MPa, respectively. Increasing the salt content, as in the case of PS7330, reduces the value further to 10 MPa, similar to the value for pure PEO. The PS7320 samples have, surprisingly, higher resistance, that dramatically drops to about 2 MPa for the PS7316 samples (Figure 7g). The deformation of the samples evolves through the formation of voids and crazing which indicate a ductile behaviour, which likely generate at the interface between different compositional variations of the amorphous region. The addition of PSf and, more importantly, LiTFSI changes the behaviour of the SPE. Increasing the PSf produces large voids (Figure 9a), while increasing the amount of LiTFSI makes the voids smaller and more spread throughout the amorphous phase (Figure 9b). For the highest concentration tested, the samples break because the amorphous phase separates easily from the spherulitic phase (Figure 9c). The samples containing a uniform amount of LiTFSI deform through large voids (200 µm) oriented in the force direction, but there are no clear connecting fibrils across the boundaries (Figure 9a), suggesting that the mechanism involved at interfaces between phases is cavitation, similarly to that reported by Zhou et al. [56]. Similar voids are observed at higher LiTFSI content (Figure 9b). For the PS7316 samples, it is noticeable that the breaking occurs at a much lower stress, and decohesion between the crystalline phase and amorphous phase is evident (Figure 9c). This is probably because the interaction between the PEO and the PSf phase is now changed, as suggested by the FT-IR. No significant trend is observed for the crystallite dimension, although a better polarizing microscopy might help in visualizing small crystallites, as the average size of the visible ones is typically 200 µm. The average Young’s modulus measured are higher than what would be expected from a binary blend as it does not follow a simple series or parallel model [57]. 

A summary of the mechanical and electrical properties of the films is reported in Table 4 below. A comparison with solid polymer electrolyte films produced by other authors is included.

### 3.6. Contact Angle

To evaluate the hydrophilicity of the films, the contact angle has been measured for all the samples (Figure 10a,b).

The measured angle of pure PEO is nearly 80° (Figure 10a), similar to the value reported by Ramires et al. [58] for PEO films. The addition of PSf, which has a similar contact angle [59], and LiTFSi, which is a highly hygroscopic salt, cause the film to be more hydrophilic, since the results show that the angle is reduced to about 68° for PS9150 and PS8250, and reaches 65° for PS7350. The reduction is related to the addition of the salt or the PEO–PSf-LiTFSI interaction since PSf concentration appears to have limited influence on the contact angle value. The addition of a higher amount of salt has no effect for PS7330 as it has the same angle as PS7350 at 65° but a further increase makes the material more hydrophilic since PS7320 and PS7316 have contact angles around 50° and 47°, respectively

## 4. Conclusions

Different blends of PEO–PSf–LiTFSI with different weight ratios of PEO–PSf and EO/Li have been prepared and characterised. 

The addition of PSf and LiTFSI improves the mechanical and electrical properties of PEO and makes it more hydrophilic;DSC suggests the presence of multiple phases, based on deductions from additional characterisation results;The best combinations are: PS9150, with 4.83 × 10^−5^ S/cm at room temperature, a maximum stress of 19.2 MPa and a Young’s modulus of 1410 MPa, and PS7330, with 1.18 × 10^−4^ S/cm at room temperature, a maximum stress of 10.02 MPa, and a Young’s modulus of 820 MPa;There is a threshold in the amount of LiTFSI that can be added to still have good mechanical properties as suggested by the PS7316 composition, which showed a brittle failure and significantly lower modulus and maximum stress compared to all other compositions.

## Figures and Tables

**Figure 1 polymers-15-02581-f001:**
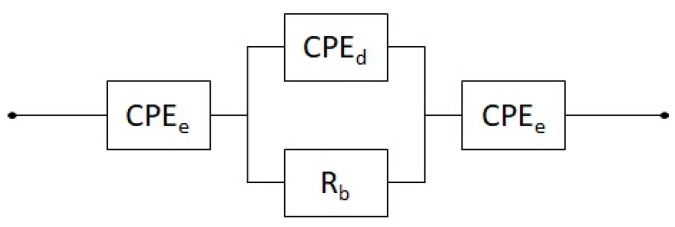
Representation of the circuit used for fitting.

**Figure 2 polymers-15-02581-f002:**
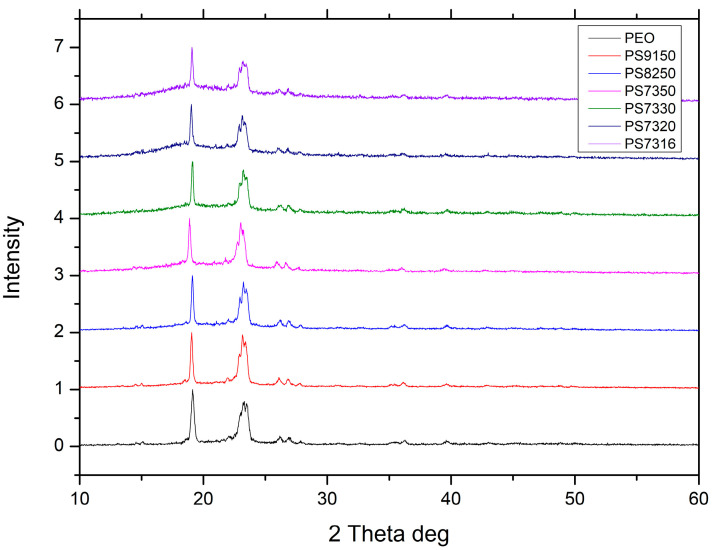
XRD scan for PEO only and all PEO–PSf compositions with offsets on the intensity scale to clarify observation.

**Figure 3 polymers-15-02581-f003:**
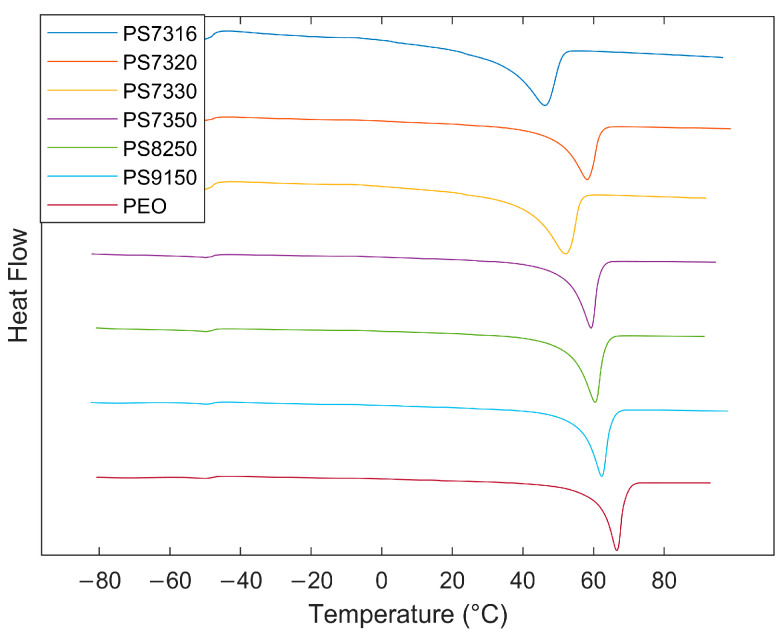
Second-heating DSC scans for PEO only and all PEO–PSF composition with offset on the vertical axis to clarify observation.

**Figure 4 polymers-15-02581-f004:**
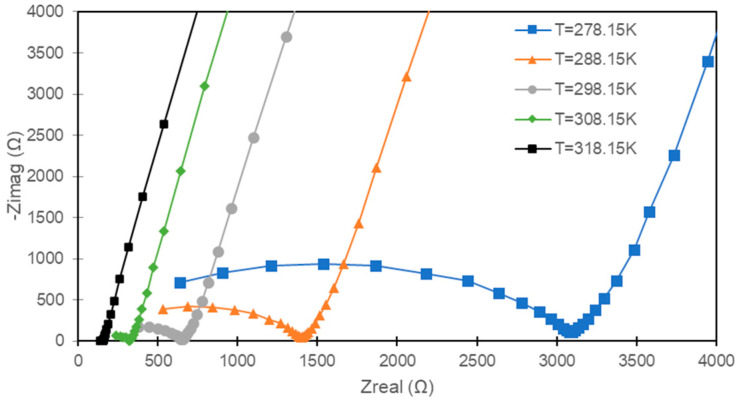
Temperature-dependent EIS curves PS7330; lines are for guidance.

**Figure 5 polymers-15-02581-f005:**
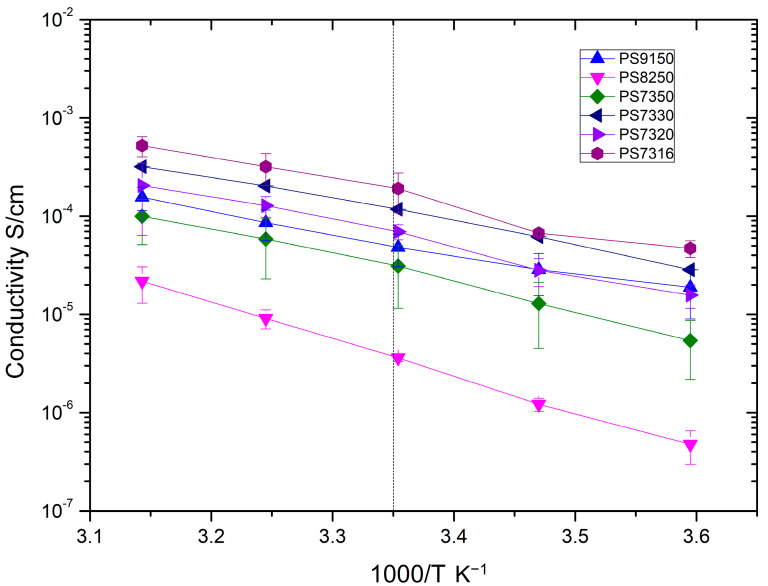
Temperature-dependent ionic conductivity of PEO–PSF samples.

**Figure 6 polymers-15-02581-f006:**
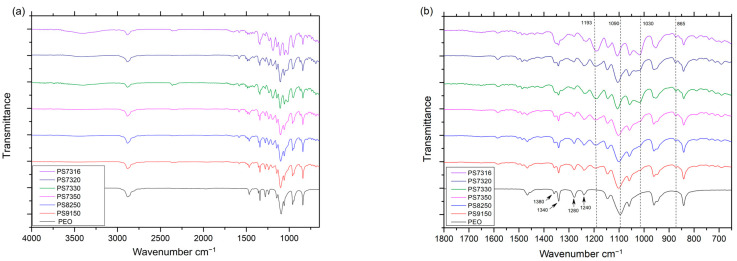
FT-IR scans for PEO and all PEO–PSf samples. (**a**) Between 4000 cm^−1^ and 650 cm^−1^; (**b**) between 1800 cm^−1^ and 650 cm^−1^.

**Figure 7 polymers-15-02581-f007:**
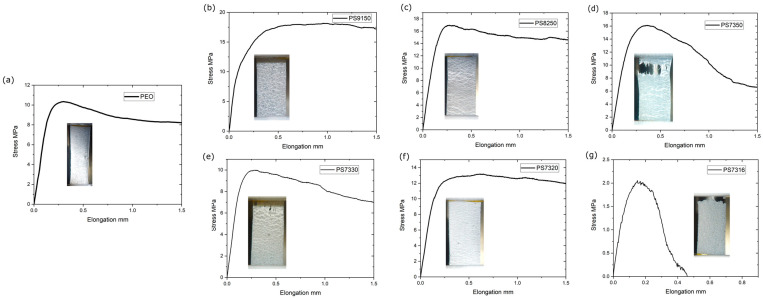
Representative stress-elongation curve for the different compositions. Images are taken at 1.5 mm deformation for all samples except for PS7316. (**a**) PEO; (**b**) PS9150; (**c**) PS8250; (**d**) PS7350; (**e**) PS7330; (**f**) PS7320; (**g**) PS7316. Images are taken using a digital camera during the test.

**Figure 8 polymers-15-02581-f008:**
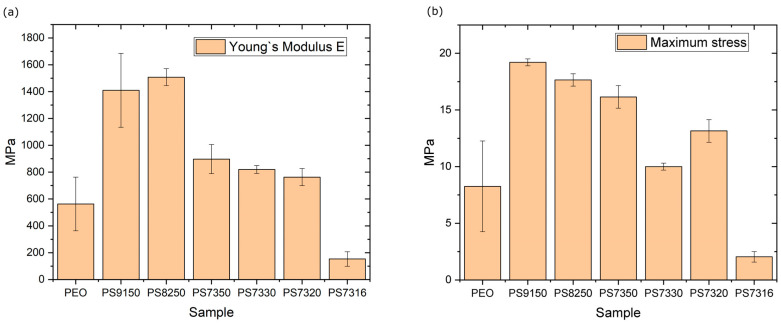
(**a**) Young’s modulus E; (**b**) maximum stress for the various composition with standard deviation.

**Figure 9 polymers-15-02581-f009:**
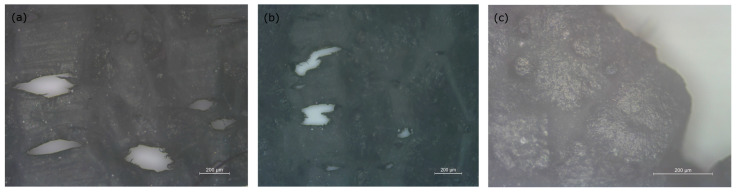
(**a**) 5× magnification of PS9150; (**b**) 5× magnification of PS7330; (**c**) 10× magnification of PS7316.

**Figure 10 polymers-15-02581-f010:**
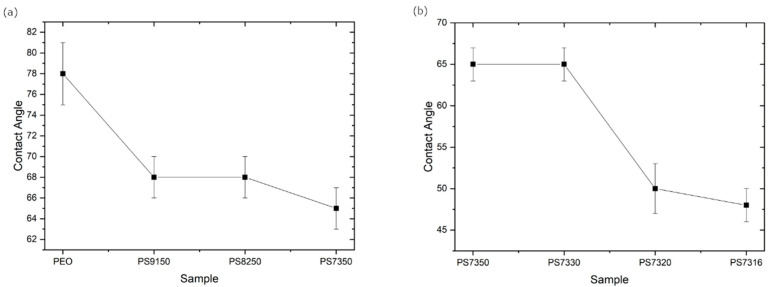
Contact angle measured for (**a**) PEO with increasing amount of PSf and fixed amount of salt and for (**b**) PS7350 with increasing amount of LiTFSI.

**Table 1 polymers-15-02581-t001:** Sample specifications and sample code names.

Samples Composition	Sample Code Names
PEO–PSf 90-10 having EO/Li = 50/1	PS9150
PEO–PSf 80-20 having EO/Li = 50/1	PS8250
PEO–PSf 70-30 having EO/Li = 50/1	PS7350
PEO–PSf 70-30 having EO/Li = 30/1	PS7330
PEO–PSf 70-30 having EO/Li = 20/1	PS7320
PEO–PSf 70-30 having EO/Li = 16/1	PS7316

**Table 2 polymers-15-02581-t002:** DSC second-heating data with XRD crystallinity comparison.

Sample	$T_{c}$, Crystallization Peak Temperature (°C)	$T_{M}$, Melting Peak Temperature (°C)	$T_{g}$, Glass Transition Temperature (°C)	${\Delta H}_{e}$, Enthalpy of Melting (J/g)	$X_{C(DSC)}$, Crystallinity DSC	$X_{C(XRD)}$, Crystallinity XRD
PEO	47.73	66.56	−47.70	130.6	63.71%	68%
PS9150	42.12	64.03	−47.76	125	60.98%	61.8%
PS8250	40.38	60.36	−47.60	102	49.76%	49.00%
PS7350	38.73	59.3	−47.64	87.15	42.51%	44.40%
PS7330	31.78	51.93	−47.69	56.79	27.70%	43.90%
PS7320	36.60	58.15	−47.71	83.93	40.94%	39.00%
PS7316	27.63	46.21	−47.84	48.05	23.44%	35.60%

**Table 3 polymers-15-02581-t003:** Summary of FT-IR peaks.

Wavenumber (cm^−1^)	Vibration Modes
840–850	CH_2_ rocking
865	Benzene ring
1030–1070	SO_2_
1090	C-O-C (crystalline)
1193	CF_3_ stretching
1240–1280	CH_2_ twisting
1340–1359	CH_3_ wagging
>3000	O-H stretching (water)

**Table 4 polymers-15-02581-t004:** Comparison between two of the films produced for this work and some of those from different authors.

SPE	EO:Li Ratio	E-RT (MPa)	$\sigma_{max}$ at RT (MPa)	*σ* at RT (S/cm)	Ref.
PS9150	50:1	1410	19.20	4.83 × 10^−5^	-
PS7330	30:1	820	10.03	1.18 × 10^−4^	-
PEO with 1% wt GO	20:1	-	1.31	1.5 × 10^−6^	[16]
PEO–PVdF with 2% wt ZnO	10:1	-	2.35	6 × 10^−5^	[23]
PEO/PSf copolymer with succinotrile	8:1	-	14.7	10^−7^	[26]
PEO–PESf–PVC	16:1	-	-	10^−5^	[27]
PEO–LiTFSI–LAGP60	20:1	103.4	2.3	2 × 10^−6^	[20]
PEO–LiBF_4_–LAGP20	20:1	400	12		[13]

## Data Availability

Data supporting this study are included within the article.

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
