# Peer review of "Electrical and Mechanical Characterisation of Poly(ethylene)oxide-Polysulfone Blend for Composite Structural Lithium Batteries"

_polymers, 2023, doi:10.3390/polym15112581_

Round 1

Reviewer 1 Report

Stefano Russo et al. Report Electrical and mechanical characterisation of  poly(ethylene)oxide-polysulfone blend for composite structural lithium batteries. The work has revealed the effect of blending of composite materials on field of lithium ion batteries. In addition, the authors investigate in detail X-ray diffraction analysis, EIS, FTIR, Micro tensile testing. The topic is very interesting.  This work is of high interest to readers working in the field of lithium ion batteries and their applications.

The authors have worked previously on PEO-based polymer blend electrolyte for composite structural battery, Polymer-Plastics Technology and Materials, 62:8, 1019-1028, DOI: 10.1080/25740881.2023.2180391.  The topic is highly exciting and the study is very informative.    

However, the clear advantage of this work compare to existing literature is missing and it requires minor revisions in order to meet the journal's requirements.

  1. SAX (small angle X-ray) will be a great addition to it. 
  2. Compare the data with the existing literature. 
  3. A clear or significant advantage of this work compare to other studies is missing. 
  4. Hydrophilic behavior of composite material is missing. Which is a very important parameter for lithium ion batteries. 

Author Response

Responses in the attached word document

Reviewer 2 Report

The authors did a fundamental study on PEO/Li salt on different molecular ratios, and realized that EO/Li ratio equals to 16 could potentially give a better results compared with other conditions, which is consistent with many published results. Given the importance of the findings, the quality and the length of the manuscript. I suggest to publish on "Polymer" after a major revision. My comments are as follow:

1. Why does the authors choose EO/Li ratio to be 16:1 instead of 15:1. 16:1 looks like an arbitrary number. 

2. The manuscript title is for "Structural" Li batteries, however, no "structural battery" discussion is mentioned in the introduction part. The following paper should give an idea on structural battery designs and is encouraged to be cited: https://doi.org/10.1016/j.ensm.2023.02.031

3. It would be much better if the authors can list a table to identify which peak correspond to what bond vibration for Figure 6. It would also be great if the authors can have time to perform Raman to cross check the bond vibration shown in Figure 6. 

4. The authors need to combine Figure 7 and Figure 8. 

5. For Figure 4, the EIS data of all other PEO-based composite compounds are missing, the authors need to include not only PS7330, but also PS9150, PS8250, PS7350, PS7320 and PS7316.

6. In Figure 2, since the authors were using LiTFSI which is air stable, it is highly encouraged to do an XRD for pure LiTFSI to the peaks with respect the PEO-based composite and did some analysis on it. 

The abstract and the entire manuscript is extremely difficult to read and lack of logics. Definitely need to do a major improvement.

Author Response

Responses in the attached document

Round 2

Reviewer 2 Report

The revised manuscript is suitable to be published in the present form. 

Needs professional edition